# Evaluation of Intraligamentous and Intraosseous Computer-Controlled Anesthetic Delivery Systems in Pediatric Dentistry: A Randomized Controlled Trial

**DOI:** 10.3390/children10010079

**Published:** 2022-12-30

**Authors:** Andrea Prol Castelo, Eliane García Mato, Iván Varela Aneiros, Lucía Sande López, Mercedes Outumuro Rial, María Teresa Abeleira Pazos, Berta Rivas Mundiña, Jacobo Limeres Posse

**Affiliations:** Medical-Surgical Dentistry Research Group (OMEQUI), Health Research Institute of Santiago de Compostela (IDIS), University of Santiago de Compostela (USC), Santiago de Compostela, 15782 Corunna, Spain

**Keywords:** local anesthesia, computer-controlled local anesthetic delivery systems, intraligamentary anesthesia, intraosseous anesthesia, pediatric dentistry

## Abstract

Computer-controlled local anesthetic delivery systems (CDS) represent one of the resources that have progressed the most in recent years, but their efficacy and applicability in pediatric dentistry is still the subject of certain controversies. This randomized, controlled, split-mouth clinical trial assessed two CDS in children (*n* = 100) with deep caries in the temporary dentition that required invasive therapeutic procedures, using inferior alveolar nerve block as the gold standard. Half of the patients (*n* = 50) underwent the intraligamentary technique (Wand STA^®^) on one side of the mouth and conventional inferior alveolar nerve block on the contralateral side, while the other half (*n* = 50) underwent the intraosseous technique (QuickSleeper^®^) on one side of the mouth and conventional inferior alveolar nerve block on the contralateral side. The following were considered covariates: age, sex, type of dental procedure and the applied local anesthesia system. The outcome variables were the pain caused by the anesthesia injection, the physical reaction during the anesthesia injection, the need for anesthetic reinforcement, pain during the therapeutic procedure, the overall behavior during the visit, the postoperative morbidity and, lastly, the patient’s preference. In conclusion, we confirmed the efficacy of intraligamentary and intraosseous techniques administered using a CDS for conducting invasive dental treatments in children, their advantages compared with inferior alveolar nerve block in terms of less pain generated by the anesthesia injection and less postoperative morbidity, as well as the pediatric patients’ preference for CDS versus conventional techniques.

## 1. Introduction

Even well into the 21st century, the association between “pain” and “dental treatment” persists in many countries, determining in large measure the attitude of the population who visit the dentist [1]. In pediatric dentistry, limiting painful procedures as much as possible and managing fear and anxiety are fundamental for obtaining an adequate degree of collaboration [2]. Of all the procedures performed in the dental office, the administration of local anesthesia (LA) is the one that patients associate most with pain and accordingly is the one that creates the most anxiety [1,3].

The pain attributed to LA is the consequence of the mechanical trauma of inserting the needle and of distending the tissues caused by the anesthetic agent [4]. Therefore, it is essential to employ proper anesthetic technique, in keeping with the patient’s age, given that it constitutes an essential determinant in the perception of the pain the patient experiences [5]. Conventional LA techniques include injection, nerve blocking and intraligamentary and intrapulpal injection; it has been suggested that some are inherently more painful than others [6].

The most commonly employed LA techniques in pediatric dentistry are periapical injection and nerve blocking, using a dental syringe, disposable carpules and conventional needles [7]. It has been suggested that, in the posterior mandibular sectors, the efficacy of the injection technique is less than that of the nerve block, as has been demonstrated when conducting pulpal treatments of temporary mandibular molars, probably because the bone density impedes the dissemination of the anesthetic agent [8]. In recent years, and with the introduction of new active ingredients such as articaine, these assertions have been questioned, given that articaine increases the efficacy of mandibular anesthesia injection [9]. However, inferior alveolar nerve block (IANB) can be difficult to achieve in certain patients, even in the absence of acute pulpitis [10], is more painful than the injection technique [11] and intraligamentary technique [12] and entails a greater risk of self-inflicted soft tissue injury [13] and of other complications such as nerve damage [14] and needle fractures [15].

Accordingly, new LA administration methods have been developed that are easier to implement and more predictable and comfortable for the patient, including computer-controlled local anesthetic delivery systems (CDS), periodontal injection techniques, needleless systems and intraseptal or intrapulpal injection [7]. The CDS developed at the end of the 1990s represent one of the resources that have progressed the most in recent years and are based on applying computer technology to control the flow of the anesthetic through the needle, modulating the pressure of the injection and the speed at which the liquid is injected into the tissues [16].

The advantages and clinical efficacy of CDS have been confirmed in pediatric dentistry, especially using the intraligamentary anesthesia technique (CDS-ILA) [17]. However, a number of authors have suggested that this system produces less pain and more effective anesthesia in adults than in children [18]. Intraosseous anesthesia (IOA) is considered a conventional technique consisting of injecting the anesthetic agent directly into the spongy periradicular bone. This technique initially had certain highly specific indications, such as nerve block failure, short-lasting procedures, situations in which residual anesthesia of the soft tissue should be avoided and patients with an increased risk of bleeding [19]. In recent decades, anesthetic injection methods have been developed that incorporate a computer-controlled administration system and bone perforation (CDS-IOA), which are considered an alternative or complement to classical injection techniques in children and adolescents [20].

The aim of this study was to assess the anesthetic efficacy of CDS-ILA and CDS-IOA systems in pediatric patients with deep caries in temporary dentition who require invasive therapeutic procedures, using IANB as the gold standard.

## 2. Materials and Methods

### 2.1. Study Design

We established a randomized, controlled, split-mouth clinical trial. Half of the patients underwent IANB during the first visit and CDS-ILA or CDS-IOA during the second, while the other half underwent CDS-ILA or CDS-IOA first and IANB second. The study was conducted in the Pediatric Dentistry Unit of the Faculty of Medicine and Dentistry of the University of Santiago de Compostela (Corunna, Spain). The study protocol was approved by the Regional Research Ethics Committee of the Government of Galicia (registration code 2016/123). The legal guardians of all of the patients completed and signed a specific informed consent form for study participation.

This article was prepared following the standards established for the presentation of information of trials (Consolidated Standards of Reporting Trials [CONSORT]) [21].

### 2.2. Study Group

The study group consisted of 100 pediatric patients who met the following inclusion criteria (Appendix A): age 6–12 years; co-operative (grade 3–4 on the Frankl behavior scale during the initial diagnostic visit) [22]; having 2 contralateral homologous deciduous mandibular molars that required “deep” to “very deep” fillings due to caries evaluated radiologically, that penetrated ≤50% and >50%, respectively, of the thickness of the dentin, or pulpotomies of vital teeth due to caries with pulp involvement, or extraction of teeth with crown destruction and at least half of the root remaining. We applied the following exclusion criteria: having had previous dental experiences; systemic disease; consumption of drugs in the previous 48 h that might change the perception of pain (e.g., analgesics); mobility of the selected molar, fistulae, radicular resorption, pulpal calcification, radiological defects (periapical or interradicular radiolucency), or diseased periodontal pockets; communication difficulties; allergy to local anesthetics; refusal to accept treatment; and not attending some of the visits.

### 2.3. Interventions

Before injecting the anesthesia, all patients were administered a topical anesthetic solution (gel) of 20% benzocaine (Hurricaine^®^, Clarben, Madrid, Spain) for 1 min at the location where the subsequent anesthesia injection was planned.

IANB was administered with a conventional Aspijet^®^ syringe (Ronvig A/S Degrees of freedom, Daugaard, Denmark) according to the classical technique. CDS-ILA was applied using a Wand STA^®^ device (Milestone Scientific, Roseland, NJ, USA), following the manufacturer’s instructions. CDS-IOA was performed using the QuickSleeper^®^ system (Dental HiTec, Cholet, France), following the manufacturer’s instructions.

The anesthetic agent employed was 2% lidocaine with epinephrine 1:100,000 (Xilonibsa^®^, Inibsa, Barcelona, Spain) in 1.8 mL carpules. We employed a short needle (25 mm/30G) and administered 1 carpule in the case of IANB. For CDS-ILA, we administered half a carpule (0.9 mL) of anesthesia in each radicular portion (mesial and distal). For CDS-IOA, we also used half a carpule (0.9 mL) of anesthesia, which was injected into the interradicular space distal to the tooth to anesthetize. All the patients were anesthetized using 2 techniques (one in each quadrant) in 2 separate sessions (IANB versus CDS-ILA or instead IANB versus CDS-IOA), always starting by the right quadrant. The anesthesia was always administered by a single operator, with more than 10 years of professional experience as a pediatric dentist and skilled in the use of all anesthesia systems employed. The outcome variables were recorded by an assessor who was previously trained and qualified by viewing 10 videos on dental procedures performed in children (intra-assessor Kappa coefficient, 0.94).

In all patients, 2 homologous deciduous molars of contralateral mandibular quadrants were treated. The therapeutic procedure began immediately after finishing the anesthetic injection in the case of CDS-ILA and CDS-IOA and once the patient confirmed numbness of the lip and tongue in the case of IANB. The interval between the 2 sessions was 7–14 days.

Each patient included in the study was assigned an identification code, and a randomization by blocks process was performed (www.random.org, accessed on 30 January 2019) to determine the administration order of the various anesthetic techniques.

### 2.4. Blinding

The operator and the patients were aware of the applied anesthetic system because the devices are highly specific and difficult to hide. In each patient, we performed 2 procedures in 2 separate sessions using 2 distinct anesthesia systems. The results from each patient were included in an anonymous registry with a code to identify the anesthesia system being used. The statistical analysis of the results was performed by an external company that was unaware of the meaning of the codes.

### 2.5. Covariates

We recorded the age (years) and sex (male/female) of all participants, the type of dental procedure (filling, pulpotomy or extraction) and the local anesthesia system applied (IANB, CDS-ILA or CDS-IOA).

### 2.6. Outcome Variables

The outcome variables were the pain caused by the anesthesia injection, the physical reaction during the anesthesia injection, the need for anesthetic reinforcement, pain during the therapeutic procedure, the overall behavior during the visit, the postoperative morbidity and the patient’s preference. The assessment of the pain sensation experienced by the patient during the administration of anesthesia was recorded after completing the injection procedure, applying the Wong–Baker visual pain scale [23] validated for pediatric patients and recoded to facilitate its analysis (0 = Doesn’t hurt at all, 1 = Hurts a little, 2 = Hurts quite a bit, 3 = Hurts a lot). We also analyzed the patients’ physical reaction during the anesthesia injection by using the FLACC (“Face, Legs, Activity, Cry, Controlability”; 0–10 points) scale, which evaluates disruptive behaviors [24].

At the end of each session, we asked the patients to indicate whether they felt pain during the therapeutic procedure, once again using the Wong–Baker scale [23]. The patients’ overall behavior during the treatment session was measured as an indirect indicator of the degree of comfort using the Frankl scale (1 = Definitely negative, 2 = Negative, 3 = Positive, 4 = Definitely positive) [22].

After finishing the 2 treatment meetings, all patients were asked as to their preferred anesthesia system.

We determined the effectiveness/failure of each anesthetic technique employed based on the perception of discomfort/pain during the treatment that required the administration of an anesthetic reinforcement applying injection techniques with the conventional system.

The patients’ legal guardians were provided a contact phone number to report any incidents that might be related to the dental treatment (discomfort, pain that required the administration of analgesics or nibbling injuries). Additionally, a member of the dental team made a follow-up phone call 24 h after the dental procedure was completed to record the postoperative morbidity.

### 2.7. Sample Size

For the sample size, we based our estimate on the computer-controlled local anesthetic delivery system that has been least often used in pediatric dentistry (CDS-IOA) and on the primary outcome variable of the study, “the pain caused by the anesthesia injection”. We hypothesized that CDS-IOA would reduce the pain generated by the anesthesia injection compared with that produced during an IANB with a conventional syringe. Accordingly, we considered the correlation induced by the paired nature of the data. Analyzing previous studies that employed a visual analog scale (VAS) to determine the pain intensity, assuming standard deviations corresponding to a VAS score of 1.2 points [25], a correlation between the VAS score for the same patient between 2 anesthesia modalities of 0.6 points and a risk of type I error of 0.05, a total of 30 patients should be treated with each anesthesia system to achieve a statistical power of 80%. In this study, we increased the sample size to 50 patients with each anesthesia system to ensure the trial’s statistical power.

### 2.8. Statistical Analysis

The tool used for performing the analysis was the free software R v 4.0.5 [26]. Given that the data are paired, we used the following statistical tests: Wilcoxon signed-rank test with continuity correction, McNemar’s chi-squared test with continuity correction and Student’s *t*-test for paired samples. *p*-values < 0.05 were considered statistically significant. We applied mixed generalized linear models that allow different types of outcome variables to be analyzed when considering correlated observations. To reduce the models’ complexity, certain variables with multiple categories (e.g., “pain due to anesthesia injection”) were recoded as binomial variables. The best model was considered that which had the lowest Akaike information criterion (AIC) value or the highest AIC difference. In the logistic regression model, the response variable was log(odds) = log (p/1 − p).

## 3. Results

The study group consisted of 100 patients (55 female and 45 male participants), with a mean age of 7.6 ± 2.0 years (range 6–12 years). The teeth involved were the first deciduous mandibular molars in 36 patients and the second molars in the remaining 64 patients. The procedures performed were deep fillings (28%), very deep fillings (30%), pulpotomies (28%) and extractions (14%).

The values for the outcome variables are listed in Table 1. Thirty-six percent of the injections with CDS-ILA were painless, compared with 28% with CDS-IOA and 7% with IANB. The lowest value for physical reaction during the anesthesia injection (FLACC score) was achieved with the CDS-ILA system. Anesthetic reinforcement was necessary for 8%–17% of the patients. The pain during the therapeutic procedure was “Doesn’t hurt at all” or “Hurts a little” (scores 0 and 1 on the Wong–Baker scale) for 94% of the patients with CDS-ILA, for 86% of those with IANB and for 84% of those with CDS-IOA. The overall behavior during the treatment session with the three anesthesia systems was “positive” or “definitely positive” (scores 3 and 4 of the Frankl scale) in 84%–92% of the patients. Fifty-two percent, twenty-eight percent and twelve percent of the patients had some postoperative complication with IANB, CDS-IOA and CDS-AFI, respectively. Eighty-two percent of the patients who underwent CDS-ILA preferred this system to conventional IANB. Among the patients who underwent CDS-IOA, 76% preferred this system to conventional IANB.

When comparing the values of the outcome variables using the CDS-ILA versus the IANB system, there were statistically significant differences in favor of CDS-ILA in the following variables: “pain due to anesthesia injection” (*p* < 0.001), “physical reaction during the anesthesia injection” (*p* < 0.001), “postoperative morbidity” (*p* < 0.001) and “type of postoperative complication” (*p* < 0.001) (Table 2).

When comparing the values of the outcome variables using the CDS-IOA versus the IANB system, there were statistically significant differences in favor of CDS-IOA in the following variables: “pain due to anesthesia injection” (*p* = 0.005), “postoperative morbidity” (*p* = 0.009) and “type of postoperative complication” (*p* = 0.014) (Table 2).

In the patients who underwent anesthesia injections with the CDS-ILA and IANB systems, the best model for predicting “pain due to anesthesia injection” exclusively included the “anesthesia system” (CDS-ILA decreased the log(odds ratio) by 1.86 compared with IANB). The best model for predicting the “patients’ physical reaction” exclusively included the “anesthesia system” (CDS-ILA decreased the coefficient estimated by the model by 1.12 compared with IANB). The best model for predicting the need for “anesthetic reinforcement” only included “age” (for each year of age, the log(odds ratio) decreased by 0.47). The best model for predicting “pain during the therapeutic procedure” only included “sex” (being female increased the log(odds ratio) by 1.34 compared with being male). We could not establish a model for predicting the “overall behavior during the treatment session” because none of the covariates were statistically significant. The best model for predicting “postoperative morbidity” included the “anesthesia system” (CDS-ILA decreased the log(odds ratio) by 2.68 compared with IANB)and the “type of dental procedure” (pulpotomy increased the log(odds ratio) by 2.45 compared with deep filling; extraction increased the log(odds ratio) by 2.67 compared with deep filling). The strength of the relationship between exposure to an anesthesia system and the outcome variables is outlined in Table 3. The strength of the relationship between exposure to a considered risk factor (covariates) and the outcome variables is outlined in Appendix A.

For the patients who underwent anesthesia injections with the CDS-IOA and IANB systems, the best model for predicting “pain due to anesthesia injection” exclusively included the “anesthesia system” (CDS-IOA decreased the log(odds ratio) by 2.23 compared with IANB). The best model for predicting the “patients’ physical reaction” included “age” (for each year of age, the log(odds ratio) decreased by 0.40) and the “anesthesia system” (CDS-IOA decreased the log(odds ratio) by 1.68 compared with IANB). We could not establish a model for predicting the variables “anesthetic reinforcement”, ”pain during the therapeutic procedure” or “overall behavior during the treatment session” because none of the covariates were statistically significant. The best model for predicting “postoperative morbidity” included the “anesthesia system” (CDS-IOA decreased the log(odds ratio) by 1.18 compared with IANB). The strength of the relationship between exposure to an anesthesia system and the outcome variables is outlined in Table 3. The strength of the relationship between exposure to a considered risk factor (covariates) and the outcome variables is outlined in Appendix A.

## 4. Discussion

This study confirmed the anesthetic efficacy of two computer-controlled local anesthetic delivery systems (CDS) for performing invasive dental treatments in children and assessed their potential advantages in terms of generated pain, behavioral reactions and postoperative morbidity. The study was designed as a randomized, controlled, split-mouth clinical trial, as has already been indicated [17]. Other authors have suggested combining the split-mouth design with that of parallel arms [27]. It has recently been suggested that, for this type of study, parallel trials might be preferable to crossed trials because the initial level of anxiety in the second appointment depends on the success of the initial surgery [28]; however, we consider that this potential bias is neutralized by the randomization process and an appropriate sample size.

The injection of the anesthetic agent with the CDS-ILA system was less painful than with IANB, confirming the results of other previously published pediatric series [29,30,31,32,33] and of a number of systematic reviews [6,17,18,34]. However, a number of authors have indicated that although the CDS-ILA system causes less pain than conventional anesthesia, the difference between the two methods does not achieve statistical significance [35,36,37], particularly when assessing studies with a low risk of bias [38]. One of the main determinants is the pain assessment tool, because the perception of pain incorporates a subjective component and has a multidimensional character [39]. It has been shown that there are no differences in the physiological parameters indicative of pain (heart rhythm and blood pressure) when CDS-ILA or conventional anesthesia is applied [6]. This study applied the Wong–Baker scale [19], a tool previously used by numerous researchers [33,37,40] and whose reliability and validity have already been demonstrated [41,42]. Our results agree with those of previous studies that applied this scale for measuring pain [17].

Applying the FLACC Scale [24], we also showed that the values reached when assessing the physical reaction during the anesthesia injection were lower with the CDS-ILA system than with IANB. This scale has already been used to assess the advantages of CDS-ILA [32] and continues to be employed as a subjective marker of pain in clinical trials of anesthetic efficacy in dentistry [43].

The success of anesthesia is an outcome variable that is not usually assessed in clinical trials on dental anesthesia in children and adolescents [28]. The anesthetic efficacy of CDS-ILA was presumably similar to that of IANB, given that when analyzing the need for anesthetic reinforcement and the patients’ perceived pain during the performance of the therapeutic procedure we observed no differences between the two systems. Studies have indicated that the efficacy of CDS-ILA is similar to that of conventional techniques for performing invasive dental procedures in children, such as pulpotomies and extractions of first molars [30,44]. Anecdotally, it has even been suggested that CDS-ILA can provide faster, longer-lasting and more consistent anesthesia than anesthesia injections with conventional syringe [45] and that the need for reinforcement can be more frequent with conventional anesthesia techniques [36].

The adverse effects constitute another outcome variable that is not usually recorded in trials on local dental anesthesia in children [28]. In this study, postoperative morbidity was less frequent with CDS-ILA than with IANB, especially because, after using CDS-ILA, none of the patients presented postoperative pain or nibbling injuries, which is one of the inherent advantages to the intraligamentary anesthesia technique [7,46].

Most of the patients expressed a preference for the CDS-ILA system over IANB, as has been previously noted in the literature, both in terms of satisfaction with the procedure [35,47] and in preference for one anesthesia system compared with the other in children who underwent restorative or pulpal treatments [47,48].

The available literature in English on the use of CDS-IOA in pediatric dentistry is highly scarce [20,21,49,50,51]. It is possible that the potential risk of damaging the neighboring dental structures (e.g., permanent tooth germs) directly by the action of the needle during perforation of the bone or by local osteonecrosis by the generated heat [52] makes dentists reluctant to use this technique on children. However, it should be emphasized that incidents with CDS-IOA are highly uncommon [53], and only a few isolated cases in adults have been reported [54,55]. In agreement with other authors, the anesthesia injection with the CDS-IOA system was less painful than conventional anesthesia [49,50,51].

The physical reaction during the anesthesia injection was similar for the CDS-IOA system and the IANB, although this is determined by the anesthesia system and patient age. To date, there have been no studies comparing the disruptive behaviors generated in children by the CDS-IOA system compared with CDS-ILA [28]. The physical appearance of the anesthesia system is important for children [56], but anxiety appears to be even more of a determinant in the perception of pain than the anesthesia device employed [57]; therefore, the use of camouflaged syringes that improve the child’s behavior and reduces their anxiety [58] could be particularly useful when the CDS-IOA system is applied.

In the present study, the postoperative pain was similar with CDS-IOA and IANB. At least one pediatric series has been published indicating that the prevalence of postoperative pain was lower when using CDS-IOA [49]. Paradoxically, in another series of adult patients, postoperative pain in the injection site was more frequent when administering intraosseous anesthesia than with conventional anesthesia [53]. In agreement with other authors, we observed no nibbling injuries with the CDS-IOA system [20].

Most of the patients indicated their preference for the CDS-IOA system over IANB, a result that confirms those of previous studies both in children [25] and in adults [59].

This study has a number of limitations that preclude generalization of the results and that are inherent to the inclusion criteria: age, degree of patient cooperation, teeth involved and type of dental procedure conducted. It has been suggested that CDS are more effective in adults than in children [18], and there are very few studies in patients younger than 4 years [60]. Recently, clinical trials of dental anesthesia with CDS in uncooperative children have also been recommended [34], given that there are very few published studies on this subject [61]. The pain caused by anesthesia injections can also be determined by the injection site and the area that will be anesthetized [62]. The efficacy of CDS in deciduous molars can vary depending on the type of dental procedure to be performed [63]. Other study limitations are that we did not assess other outcome variables, such as the time needed to achieve the anesthetic effect and its duration, and we did not determine the efficacy of the system with more specific tools [64].

## 5. Conclusions

Assuming the limitations of the present study, we confirmed the efficacy of local intraligamentary and intraosseous anesthesia techniques administered using a computer-controlled delivery system for performing invasive dental treatments in children. We also confirmed their advantages compared with inferior alveolar nerve block in terms of pain generated by the anesthesia injection and postoperative morbidity, as well as the pediatric patients’ preference for these anesthesia systems versus conventional techniques.

## Figures and Tables

**Table 1 children-10-00079-t001:** Percentage distribution of the scores corresponding to the outcome variables (first column), obtained from the study group according to the applied anesthesia system.

		IANB (*n* = 100)	CDS-ILA (*n* = 50)	CDS-IOA (*n* = 50)
**Pain by anesthesia injection** (Wong–Baker score)	Score 0	7%	36%	28%
	Score 1	49%	54%	52%
	Score 2	34%	10%	16%
	Score 3	10%	0%	4%
**Physical reaction during the anesthesia injection** (FLACC score)		4.4 ± 2.8 points (range 0–10.0)	2.4 ± 1.7 points (range 0–9.0)	3.8± 2.1 points (range 0–8.0)
**Anesthetic reinforcement**		17%	14%	8%
**Pain during the therapeutic procedure** (Wong–Baker score)	Score 0	50%	58%	56%
	Score 1	36%	36%	28%
	Score 2	11%	0%	16%
	Score 3	3%	6%	0%
**Overall behavior during the visit** (Frankl scale)	Score 1	1%	2%	0%
	Score 2	8%	14%	8%
	Score 3	35%	22%	28%
	Score 4	56%	62%	64%
**Postoperative morbidity**	Total	52%	12%	28%
**Type of postoperative complication**	Discomfort	28%	12%	12%
	Pain	12%	0%	16%
	Nibbling injury	12%	0%	0%

IANB, inferior alveolar nerve block; CDS-ILA, computer delivery system—intraligamentary anesthesia; CDS-IOA, computer delivery system—intraosseous anesthesia; FLACC, Face, Legs, Activity, Cry, Controllability.

**Table 2 children-10-00079-t002:** Statistical significance of the differences between the values of the outcome variables (first column), obtained from the study group according to the applied anesthesia system.

	CDS-ILA vs. IANB	CDS-IOA vs. IANB
**Pain by anesthesia injection** (Wong–Baker score)	*p* < 0.001 ^a^	*p* = 0.005 ^a^
**Pain due to the anesthesia injection** (recoded)	*p* = 0.004 ^b^	*p* = 0.003 ^b^
**Physical reaction during the anesthesia injection** (FLACC score)	*p* < 0.001 ^c^	*p* = 0.103 ^c^
**Anesthetic reinforcement**	*p* = 0.248 ^b^	*p* = 0.157 ^b^
**Pain during the therapeutic procedure** (Wong–Baker score)	*p* = 0.859 ^a^	*p* = 0.969 ^a^
**Pain during the therapeutic procedure** (recoded)	*p* = 0.803 ^b^	*p* = 0.987 ^a^
**Overall behavior during the visit** (Frankl scale)	*p* = 1.000 ^a^	*p* = 0.564 ^a^
**Behavior during the anesthesia injection** (recoded)	*p* = 0.773 ^b^	*p* = 0.439 ^b^
**Postoperative morbidity**	*p* < 0.001 ^b^	*p* = 0.009 ^b^
**Type of postoperative complication**	*p* < 0.001 ^a^	*p* = 0.014 ^a^

CDS-ILA, computer delivery system—intraligamentary anesthesia; IANB, inferior alveolar nerve block; CDS-IOA, computer delivery system—intraosseous anesthesia; ^a^ Wilcoxon signed-rank test with continuity correction; ^b^ McNemar’s chi-squared Test with continuity correction; ^c^ Student’s *t*-test.

**Table 3 children-10-00079-t003:** Computer delivery system—intraligamentary anesthesia (CDS-ILA) or computer delivery system—intraosseous anesthesia (CDS-IOA) versus inferior alveolar nerve block with conventional syringe (IANB). Strength of the relationship in the proposed model between exposure to an anesthesia system and the outcome variables.

	Pain Due to the Anesthesia Injection ^a^	Physical Reaction during the Anesthesia Injection ^b^	Anesthetic Reinforcement	Pain during the Therapeutic Procedure ^a^
**CDS-ILA versus IANB**	OR = 6.410 CI = 1.723–23.817 *p* = 0.010	EC = 1.125 SE = 0.222 *p* < 0.001	ns	ns
**CDS-IOA versus IANB**	OR = 0.107 CI = 0.012–0.950 *p* = 0.045	EC = −1.680 SE = 0.639 *p* = 0.011	ns	ns
	**Overall behavior during the visit ^c,d^**	**Postoperative morbidity**	**Type of postoperative complication ^e^**	
**CDS-ILA versus IANB**	ns	OR = 14.661 CI = 2.697–79.903 *p* < 0.001	OR = 3.980 CI = 0.991–15.957 *p* = 0.021	
**CDS-IOA versus IANB**	ns	OR = 0.306 CI = 0.094–0.992 *p* = 0.048	OR = 1.473 CI = 0.658–1.894 *p* = 0.043	

^a^ Recoded Wong–Baker score (binomial); ^b^ FLACC score (binomial); OR, Odds ratio; ns, not significant; CI, confidence interval; *p*, *p*-value; ; EC, estimated coefficient by the model; SE, standard error; ^c^ Frankl scale; ^d^ The data correspond to the extreme responses (“highly negative” versus “highly positive”); ^e^ The model is only reliable regarding “discomfort” versus “without complications” (not applicable with relation to “pain” or to “nibbling injuries”).

## Data Availability

The data are available upon request to the corresponding author.

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
