# Peer review of "Evaluation of Intraligamentous and Intraosseous Computer-Controlled Anesthetic Delivery Systems in Pediatric Dentistry: A Randomized Controlled Trial"

_children, 2022, doi:10.3390/children10010079_

Round 1
Reviewer 1 Report
1. The title of this article is "Evaluation of intraligamentous and intraosseous computer-controlled anesthetic delivery systems versus conventional inferior alveolar nerve block in pediatric dentistry". Thus, the number of groups should be 3; intraligamentous, intraosseous, and conventional inferior alveolar nerve block". However, this was split mouth clinical trials and the number of groups was 2. It is confusing.
2. According to the abstract, the groups are one with an intraligamentary technique (Wand STA® ) and the other with an intraosseous technique (QuickSleeper®). Thus, any comparison should be done in these groups. However, conclusion was not matched to this study design.
3. In the method section, authors stated differently to the abstract. It was stated that "Half of the patients underwent IANB during the first visit and CDS-ILA or CDS-IOA during the second, while the other half underwent CDS-ILA or CDS-IOA first and IANB second." In this case, authors already assumed that the efficacy of CDS-ILA and CDS-IOA is the same. Why did authors use these two techniques as mixed pattern? If they have the same effect, only one of them should be used and compared to conventional one.
4. How could authors define this study design as split-mouth clinical trials? According to the method, the difference was only the timing of the anesthesia. There was no split-mouth design.
5. When authors determine the sample size of this study, the hypothesis of this study should be presented. Then, reviewer can assess whether their method was correct or not.
6. In case of pediatric patients, the anxiety level can influence on overall attitude in the clinics. I'm not sure whether Wong-Baker score is reliable in the evaluation of the pediatric patients.
Author Response
Reviewer 1
- The title of this article is "Evaluation of intraligamentous and intraosseous computer-controlled anesthetic delivery systems versus conventional inferior alveolar nerve block in pediatric dentistry". Thus, the number of groups should be 3; intraligamentous, intraosseous, and conventional inferior alveolar nerve block". However, this was split mouth clinical trials and the number of groups was 2. It is confusing.
Response
We initially included “versus conventional inferior alveolar nerve block” in the title to indicate that we had used a gold standard as a reference. Based on the reviewer’s comments and to avoid confusion, we have changed the manuscript’s title as follows: “Evaluation of intraligamentous and intraosseous computer-controlled anesthetic delivery systems in pediatric dentistry”.
- According to the abstract, the groups are one with an intraligamentary technique (Wand STA® ) and the other with an intraosseous technique (QuickSleeper®). Thus, any comparison should be done in these groups. However, conclusion was not matched to this study design.
Response
The reviewer’s interpretation is correct; however, as specified in the Abstract “using inferior alveolar nerve block as the gold standard”, half (n=50) of the patients underwent the intraligamentary technique (Wand STA®) on one side of the mouth and conventional inferior alveolar nerve block on the contralateral side, while the other half (n=50) underwent the intraosseous technique (QuickSleeper®) on one side of the mouth and conventional inferior alveolar nerve block on the contralateral side. Accordingly, we have changed the Abstract to clarify this point.
With this study design, we did not consider it appropriate to compare Wand STA® versus QuickSleeper®, given that we would be assessing both techniques in different patients. To compare the two techniques, we propose performing a future split-mouth trial in which Wand STA® is applied to one side of the mouth and QuickSleeper® is applied to the contralateral side.
- In the method section, authors stated differently to the abstract. It was stated that "Half of the patients underwent IANB during the first visit and CDS-ILA or CDS-IOA during the second, while the other half underwent CDS-ILA or CDS-IOA first and IANB second." In this case, authors already assumed that the efficacy of CDS-ILA and CDS-IOA is the same. Why did authors use these two techniques as mixed pattern? If they have the same effect, only one of them should be used and compared to conventional one.
Response
The reviewer is correct, and this wording can cause confusion. As we explained in the previous response, we never use the two techniques as a mixed pattern. To clarify this point, we have included a participant flow diagram as supplemental material (Figure S1).
- How could authors define this study design as split-mouth clinical trials? According to the method, the difference was only the timing of the anesthesia. There was no split-mouth design.
Response
As we explained in response 2, the study design was definitely a split-mouth clinical trial.
- When authors determine the sample size of this study, the hypothesis of this study should be presented. Then, reviewer can assess whether their method was correct or not.
Response
The hypothesis of this study is that both the intraligamentary anesthesia (using a Wand STA®) device and the intraosseous anesthesia (using the QuickSleeper® system) can reduce the pain generated by the anesthesia injection compared with that produced during an inferior alveolar nerve block with a conventional syringe.
This comment has been added to the manuscript.
- In case of pediatric patients, the anxiety level can influence on overall attitude in the clinics. I'm not sure whether Wong-Baker score is reliable in the evaluation of the pediatric patients.
Response
As indicated in the Discussion section, the Wong-Baker scale is a tool whose reliability and validity have been demonstrated (Hicks et al., 2001; Howard et al., 2007). This scale has already been used in other studies on dental anesthesia with designs similar to ours (Langthasa et al., 2012; Giannetti et al., 2018; El Hachem et al., 2019). The Faces Pain Scale has been shown to be appropriate for assessing acute pain intensity in children aged 4–5 years or older (Hicks et al., 2001). The Wong-Baker Scale is commonly employed in children to analyze the anxiety caused by the dental environment [Bagattoni, et al. Healthcare (Basel). 2022], as well as by other invasive extraoral medical procedures (Halal Mehdi Alfatavi et al. Compr Child Adolesc Nurs. 2022).
Reviewer 2 Report
A review for Children :
Re” Evaluation of intraligamentous and intraosseous computer-controlled anesthetic delivery systems versus conventional 3 inferior alveolar nerve block in pediatric dentistry”
This is an interesting manuscript. However, the reporting of the results should be improved. Furthermore, there are other points that should be addressed.
Title: Add “ A randomized controlled trial” at the end of the Title as recommended by the CONSORT checklist, please.
Introduction:
L72: Split “ conventionaltechnique “, please.
Methods:
L141: Add more information regarding the allocation concealment mechanism and implementation, please.
L147: I would change “Potential determinants” to “Covariates”.
L151: Specify the outcome variables at the beginning of the paragraph, then elaborate on each one, please.
Results:
I strongly suggest adding a participant flow diagram.
I find tables 3 and 4 hard to follow and might be confusing for readers. I suggest only reporting the results of the Anaesthesia system “ As it is the main focus of the manuscript” and omitting the covariates “ Age, gender and dental procedure” from the table and reporting its findings in the text.
Author Response
Reviewer 2
- Title: Add “ A randomized controlled trial” at the end of the Title as recommended by the CONSORT checklist, please.
Response
Combining the reviewers’ recommendations, we propose the following as the definitive title: “Evaluation of intraligamentous and intraosseous computer-controlled anesthetic delivery systems in pediatric dentistry. A randomized controlled trial”.
- Introduction:
L72: Split “ conventionaltechnique “, please.
Response
Done
- Methods:
L141: Add more information regarding the allocation concealment mechanism and implementation, please.
Response
In each patient, we performed 2 procedures in 2 separate sessions using 2 distinct anesthesia systems. The results from each patient were included in an anonymous registry and with a code to identify the anesthesia system being used. The statistical analysis of the results was performed by an external company that was unaware of the meaning of the codes.
This comment has been added to the manuscript.
- L147: I would change “Potential determinants” to “Covariates”.
Response
“Potential determinants” has been replaced with “Covariates” throughout the manuscript.
- L151: Specify the outcome variables at the beginning of the paragraph, then elaborate on each one, please.
Response
The outcome variables were the pain caused by the anesthesia injection, the physical reaction during the anesthesia injection, the need for anesthetic reinforcement, pain during the therapeutic procedure, the overall behavior during the visit, the postoperative morbidity and the patient’s preference.
Following the reviewer’s suggestion, this paragraph has been included in the manuscript.
- Results:
I strongly suggest adding a participant flow diagram.
Response
A participant flow diagram has been included as supplemental material (Figure S1).
- I find tables 3 and 4 hard to follow and might be confusing for readers. I suggest only reporting the results of the Anaesthesia system “ As it is the main focus of the manuscript” and omitting the covariates “ Age, gender and dental procedure” from the table and reporting its findings in the text.
Response
A new table (Table 3) has been prepared, which includes only the values corresponding to the “anesthesia system” covariate.
Tables 3 and 4 of the first version of the manuscript have been included as supplemental material (Tables S1 and S2), and the main results of the covariates “age, gender and dental procedure” are reported in the text.
Round 2
Reviewer 1 Report
Most concerns raised by reviewer have been clarified except for one. In determining sample size, authors assumed that "both the CDS-ILA and the CDS-IOA can reduce the pain generated by the anesthesia injection compared with that produced during an IANB with a conventional syringe". It looks like bio-equivalence test. If authors wanted to claim both the CDS-ILA and the CDS-IOA have similar effect to IANB, bio-equivalence margin should be presented. If p-value was <0.05 in this test, it should be interpreted as two compared methods had similar effect. Hypothesis should be checked again.
Author Response
The reviewer is correct, and this wording can cause confusion. This paragraph has rewritten as follows:
For the sample size, we based our estimate on the computer-controlled local anesthetic delivery system that has been least often used in pediatric dentistry (CDS-IOA) and on the primary outcome variable of the study, "the pain caused by the anesthesia injection". We hypothesized that CDS-IOA would reduce the pain generated by the anesthesia injection compared with that produced during an IANB with a conventional syringe. Accordingly, we considered the correlation induced by the paired nature of the data.
Reviewer 2 Report
I would like to thank the authors for incorporating my comments within the revised version of the manuscript. I have no further comments to add.
Author Response
Thank you very much for your kind comment.